# ARE GENERATIVE CLASSIFIERS MORE ROBUST TO ADVERSARIAL ATTACKS?

**Yingzhen Li**
University of Cambridge, UK
`yl494@cam.ac.uk`

## ABSTRACT

There is a rising interest in studying the robustness of deep neural network classifiers against adversaries, with both advanced attack and defence techniques being actively developed. However, most recent work focuses on *discriminative* classifiers which only models the conditional distribution of the labels given the inputs. In this abstract we propose *deep Bayes classifier* that improves the classical naive Bayes with deep generative models, and verifies its robustness against a number of existing attacks. Our initial results on MNIST suggest that deep Bayes classifiers might be more robust when compared with deep discriminative classifiers.

## 1 INTRODUCTION

Deep neural networks have been shown to be vulnerable to adversarial attacks Szegedy et al. (2013); Goodfellow et al. (2014). Since then, many researchers have proposed adversarial attack and defence mechanisms, and some notable developments include: Goodfellow et al. (2014); Moosavi Dezfooli et al. (2016); Papernot et al. (2016); Carlini & Wagner (2017a); Kurakin et al. (2016); Madry et al. (2018) for attacks, and Szegedy et al. (2013); Gu & Rigazio (2014); Grosse et al. (2017); Li & Gal (2017); Feinman et al. (2017); Louizos & Welling (2017); Song et al. (2018); Madry et al. (2018) for defences. These developments enable better understanding of the robustness of deep neural networks as *discriminative classifiers* against adversaries.

Surprisingly, much less recent work has investigated the robustness of *generative classifiers* against adversarial attacks for multi-class classification, where such classifiers explicitly model the conditional distribution of the inputs given labels. In formula, denote the random variables of the input and label as $\boldsymbol{x} \in \mathbb{R}^D$ and $\boldsymbol{y} \in \{\boldsymbol{y}_c | c = 1, ..., C\}$ where $\boldsymbol{y}_c$ denotes the one-hot encoding vector for class $c$. A generative classifier first builds a *conditional generative model* $p(\boldsymbol{x}|\boldsymbol{y})$, then, in prediction time, predicts the label of a test input $\boldsymbol{x}^*$ using Bayes' rule

$$p(\boldsymbol{y}^*|\boldsymbol{x}^*) = \frac{p(\boldsymbol{x}^*|\boldsymbol{y}^*)p(\boldsymbol{y}^*)}{p(\boldsymbol{x}^*)}. \tag{1}$$

Perhaps the *naive Bayes* classifier is the most well-known generative classifier, which assumes a factorised distribution for the conditional generator, i.e. $p(\boldsymbol{x}|\boldsymbol{y}) = \prod_{d=1}^{D} p(x_d|\boldsymbol{y})$. However naive Bayes is less suitable for e.g. image and speech data, where the factorisation assumption is inappropriate. Fortunately, we can leverage the recent advances of generative modelling and apply a deep generative model for the conditional distribution $p(\boldsymbol{x}|\boldsymbol{y})$. We refer to such generative classifiers that use deep generative models as *deep Bayes* classifiers.

As an example, we use a deep latent Gaussian model (Rezende et al., 2014) which reads

$$p(\boldsymbol{x}, \boldsymbol{z}|\boldsymbol{y}) = p(\boldsymbol{x}|\boldsymbol{z}, \boldsymbol{y})p(\boldsymbol{z}), \quad p(\boldsymbol{z}) = \mathcal{N}(\boldsymbol{z}; \boldsymbol{0}, \mathbf{I}), \quad p(\boldsymbol{x}|\boldsymbol{z}, \boldsymbol{y}) = \prod_{d=1}^{D} p(x_d|\boldsymbol{z}, \boldsymbol{y}), \tag{2}$$

where $p(x_d|\boldsymbol{z}, \boldsymbol{y})$ can be Gaussian or Bernoulli distributions with parameters determined by a deep neural network talking both $\boldsymbol{z}$ and $\boldsymbol{y}$ as inputs. Importantly, this leads to a *non-factorised* conditional distribution $p(\boldsymbol{x}|\boldsymbol{y}) = \int_{\boldsymbol{z}} p(\boldsymbol{x}|\boldsymbol{z}, \boldsymbol{y})p(\boldsymbol{z})d\boldsymbol{z}$. However this marginal likelihood is intractable, and instead we use the variational auto-encoder (VAE) algorithm (Kingma & Welling, 2013; Rezende et al., 2014) to train the conditional generative model, together with an inference network $q(\boldsymbol{z}|\boldsymbol{x}, \boldsymbol{y})$

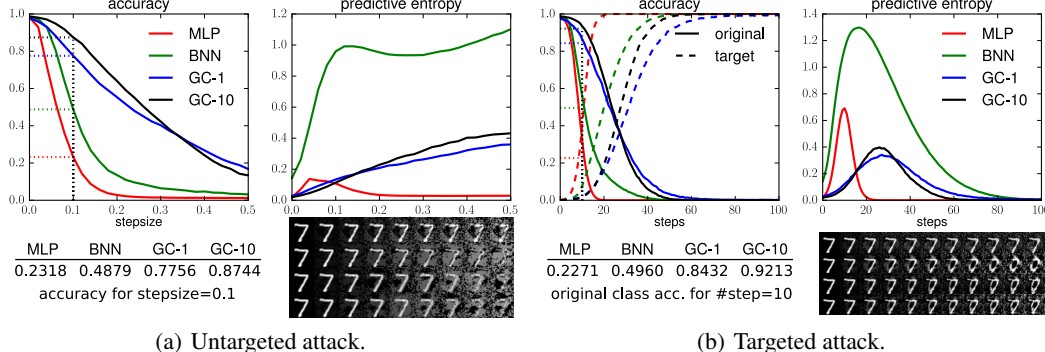

(a) Untargeted attack.            (b) Targeted attack.

Figure 1: FGSM attacks on MNIST classifiers. The predictive entropy is defined by the entropy of the classifier's output probability vector. The BNN results are taken from Li & Gal (2017).

that is also conditioned on the label $\boldsymbol{y}$. After training the predicted class probability vector $\boldsymbol{y}^*$ for a future input $\boldsymbol{x}^*$ is (approximately) computed by Bayes' rule:

$$p(\boldsymbol{y}^*|\boldsymbol{x}^*) \approx \text{softmax}_{c=1}^{C}\left(\log \sum_{k=1}^{K} \frac{p(\boldsymbol{x}^*, \boldsymbol{z}_c^k, \boldsymbol{y}_c)}{q(\boldsymbol{z}_c^k|\boldsymbol{x}^*, \boldsymbol{y}_c)}\right), \quad \boldsymbol{z}_c^k \sim q(\boldsymbol{z}|\boldsymbol{x}^*, \boldsymbol{y}_c). \tag{3}$$

where $\text{softmax}_{c=1}^{C}$ denotes the softmax operator over the $c$ axis. Therefore the output probability vector is computed in an analogous way to many deep discriminative classifiers that use softmax activation in the last layer, so that many existing attacks can be tested directly.

## 2 Initial Experimental Study

We carry out an initial test on the proposed generative classifier (GC) using VAE (3). All the attacks are taken from the CleverHans 2.0 library (Nicolas Papernot, 2017).

**MLP experiments on MNIST.** We follow Li & Gal (2017) and consider adversarial attacks on classifiers based on MLPs. Four models are tested: a normal discriminative classifier parameterised by an MLP, a Bayesian MLP network (BNN) trained with dropout rate 0.5 and tested with $K = 10$ times MC-dropout, and finally the deep Bayes classifier (trained with $\ell_2$ loss) using $K = 1$ (GC-1) and $K = 10$ samples (GC-10), respectively.

We first consider the untargeted single-step FGSM attack (Goodfellow et al., 2014) and vary the stepsize between 0.0 and 0.5. In Figure 1(a) we show the classification accuracy on the adversarial examples and also the predictive entropy measure $\mathbb{H}[\boldsymbol{y}^*]$. It is clear that the deep Bayes classifiers are most robust, where increasing $K$ also improves the test accuracy. The predictive entropy, used as a measure of uncertainty, also increases for the deep Bayes classifiers, which is as expected since the inputs are driven away from the data manifold.

We also apply the iterative version of targeted FGSM for 100 iterations with step-size 0.01. Results are shown in Figure 1(b). While this attack is more effective in terms of accuracy, again the deep Bayes models achieve the best robustness against it. Also, running this iterative attack produces a smooth interpolation between digits of the original and adversarial classes, and the predictive entropy of the classifier increases then decreases along the gradient descent path.

**CNN experiments on MNIST.** We also apply adversarial attacks to classifiers based on CNNs, and in this case we focus on the comparisons between discriminative classifiers and generative classifiers. The attack is the Carlini & Wagner $\ell_2$ attack (CW-$\ell_2$) (Carlini & Wagner, 2017a) with recommended parameters.[1] Since attacking generative classifiers take significantly longer computation time, in this initial experiment we sample 200 test images from the MNIST dataset, and craft

---

[1]https://github.com/carlini/nn_robust_attacks/blob/master/l2_attack.py

Table 1: Targeted CW-$\ell_2$ attack on CNN-based models. Here error (adv) reports the error of the adversarial inputs to the original classes, and accuracy (adv) reports the accuracy to the target labels.

|  | accuracy (clean) | error (adv) | accuracy (adv) | distortion (adv) |
|---|---|---|---|---|
| CNN | 100.00% | 99.89% | 99.89% | 1.993 |
| GC-1 | 98.77% | 15.61% | 2.65% | 2.204 |
| GC-10 | 99.15% | 20.46% | 5.86% | 2.266 |

adversarial examples targeting the classes other than the ground-truth label. This results in 1,800 attempts in total. Results are reported in Table 1, where the average distortion in $\ell_2$ distance is computed on the successful attacks. Presumably the best performance of GC-1 is due to the randomness of the classifier (3). However, we believe this randomness effect is largely removed in the GC-10 experiment, since no significant accuracy improvement is observed on clean inputs for $K > 10$. In summary, the results indicate that the deep Bayes classifier is significantly more robust to the CW-$\ell_2$ attack than the CNN baseline.

**Detecting adversarial attacks with conditional generative models.** Given an input $x$, a trained conditional generative model can produce a "reconstruction" $r(x, y)$ of $x$ conditioned on a given label $y$ (by auto-encoding or optimisation). We conjecture that, if a classifier returns an incorrect label $y^{\text{pred}} \neq y^{\text{true}}$ on $x$, then under some distance measure we can show that $d(x, r(x, y^{\text{true}})) < d(x, r(x, y^{\text{pred}}))$. Consequently, if an adversarial image of a cat is incorrectly labelled as "dog", then the "reconstructed" image will be close to an image of a dog, which is far away from the manifold of "cats" in an appropriately selected distance. We used $\ell_2$ distance as the distance measure in the appendix experiments, and confirmed our conjecture on all the attacks tested.

## 3 DISCUSSION

We have shown initial evidence that generative classifiers might be more robust to existing attacks than discriminative classifiers. The results are not conclusive as Carlini & Wagner (2017b) suggested that MNIST properties might not hold on e.g. Cifar-10. Future work will investigate deep Bayes classifiers based on auto-regressive models such as the PixelCNN (van den Oord et al., 2016b;a), and test deep Bayes classifiers on other natural image datasets such as Cifar-10 and SVHN.

Our positive results might be due to gradient masking (Papernot et al., 2017) and future work will investigate it in more detail. But we also note that many recent attacks are designed for discriminative classifiers, while many benchmark datasets have some anti-causal structures (Schölkopf et al., 2012). Consider MNIST as an example: a person first intends to write a digit ($y \sim p_\mathcal{D}(y)$), then this intention causes a writing action producing an image of that digit ($x \sim p_\mathcal{D}(x|y)$). Therefore a deep Bayes classifier is more suitable to MNIST, and it will be more robust if the deep generative model is very powerful to approximate the data distribution $p_\mathcal{D}(x|y)$.

We do not intend to claim that generative classifiers are robust to *all* possible attacks. Indeed, naive Bayes as a standard approach for spam filtering is fragile (Dalvi et al., 2004; Huang et al., 2011), and very recently Tabacof et al. (2016); Kos et al. (2017); Creswell et al. (2017) also designed attacks for (unconditional) VAE-type models. However, Dalvi et al. (2004) also showed that generative classifiers can be made more secure if aware of the attack strategy, and Biggio et al. (2011; 2014) further improved naive Bayes' robustness by modelling the conditional distribution of the adversarial inputs. This is similar to adversarial training of deep discriminative classifiers, and efficient ways for doing so with deep Bayes classifiers can be an interesting research direction.

In general, deep Bayes classifiers are less accurate than deep discriminative classifiers on classifying legitimate inputs. Also they are much more computationally expensive, limiting their applications to big neural networks and large-scale datasets such as the ImageNet. Still, a careful study of generative classifiers can inspire better designs of attack, defence and detection techniques for discriminative classifiers that use generative models as auxiliaries. Indeed Gu & Rigazio (2014); Song et al. (2018) proposed defence techniques by "purifying" adversarial inputs with auto-encoders/generative models, which moves the adversarial images towards the data manifold. Also the proposed detection method using conditional generative models has shown promising results.

ACKNOWLEDGMENTS

I thank John Bradshaw for discussions and feedback on this abstract.

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

# A  DETECTING ADVERSARIAL ATTACKS WITH CONDITIONAL GENERATIVE MODELS

We describe the detection algorithm in more detail, with the conditional VAE as an example. Given an input $x$ and a label $y$, we can "reconstruct" $x$ by

$$z \sim q(z|x, y), \quad \hat{x} \sim p(x|z, y).$$

Therefore $\hat{x}$ depends on $x$ and $y$. In experiments we do not perform sampling and instead compute the reconstruction directly:

$$r(x, y) = \mu_x(\mu_z(x, y), y), \quad \mu_z(x, y) = \mathbb{E}_{q(z|x,y)}[z], \quad \mu_x(z, y) = \mathbb{E}_{p(x|z,y)}[x].$$

Other reconstruction methods apply, e.g. one can select a distance measure $d(\cdot, \cdot)$ and define

$$r(x, y) = \arg\min_{\hat{x}} -\log p(\hat{x}|y) + \lambda d(x, \hat{x}).$$

This proposal is not investigated here and we leave it to future work.

Our conjecture is that, for an input-label pair $(x, y)$ and its adversarial pair $(x^{\mathrm{adv}}, y^{\mathrm{adv}})$ (here $y \neq y^{\mathrm{adv}}$), we can measure the distance between the input and the reconstruction, and have

$$d(x, r(x, y)) < d(x^{\mathrm{adv}}, r(x^{\mathrm{adv}}, y^{\mathrm{adv}})).$$

Therefore, a simple detection method would first compute $\bar{d}_{\mathcal{D}} = \mathbb{E}_{(x,y)\sim\mathcal{D}}[d(x, r(x, y))]$, then determine an input $x^*$ as an adversarial example for a classifier $F$ if $d(x^*, r(x^*, F(x^*))) > \bar{d}_{\mathcal{D}}$. It is also possible to have different threshold $\bar{d}_c$ for different classes, however this is not investigated here.

To verify the conjecture we perform detection tests on MNIST with a trained CNN classifier as the victim model and a conditional VAE as the generative model. The attacks in consideration are (untargeted) CW-$\ell_2$ (Carlini & Wagner, 2017a) and DeepFool (Moosavi Dezfooli et al., 2016). Table 2 reports the average distortion of successful attacks, the $\ell_2$ distance between inputs and reconstructed images, and the detection rate using the average distance $\bar{d}_{\mathcal{D}}$ as the threshold. It is clear that for both attacks $d(x^*, r(x^*, F(x^*))) > \bar{d}_{\mathcal{D}}$ in average, and the detection method is very effective. We also visualise the reconstructed images in Figure 2, where visually the reconstructions of the adversarial images look similar to images in the adversarial classes.

Table 2: Detection experiments with the conditional generative model.

| attack | distortion | distance (clean) | distance (adv) | detection rate |
|---|---|---|---|---|
| DeepFool | $1.745 \pm 0.732$ | $2.948 \pm 0.828$ | $4.930 \pm 1.150$ | 97.71% |
| CW-$\ell_2$ | $1.370 \pm 0.530$ | $2.948 \pm 0.828$ | $4.995 \pm 1.212$ | 97.52% |

# B  MODEL ARCHITECTURES

**MLP:**  The MLP has 3 hidden layers of 500 units. We use ReLU activations.

**VAE-MLP:**  The decoder takes $(z, y)$ as input and produces $x$ using a two hidden-layer MLP with hidden layer size 500. The encoder has a symmetric architecture except that it takes $(x, y)$ as inputs and return the mean and variance parameters of $q(z|x, y)$. Here $z$ is 32 dimensions.

**CNN:**  We used 4 convolutional layers with filter size 3 and 128 channels, each followed by a max-pooling operation. Then the output is fed into a one hidden-layer MLP with 500 hidden units to produce the class probability vector.

**VAE-CNN:**  The decoder takes $(z, y)$ as input and produces $x$ by a one hidden-layer MLP with 500 units, followed by a 3-layer deconvolutional neural network with filter size $3 \times 3$ and number of channels 64, 64, 1. The encoder has almost identical architecture, except that the convolutional part only takes $x$ as inputs, and the MLP part takes $y$ and the convolutional features of $x$. The latent dimension is set to $\dim(z) = 32$.

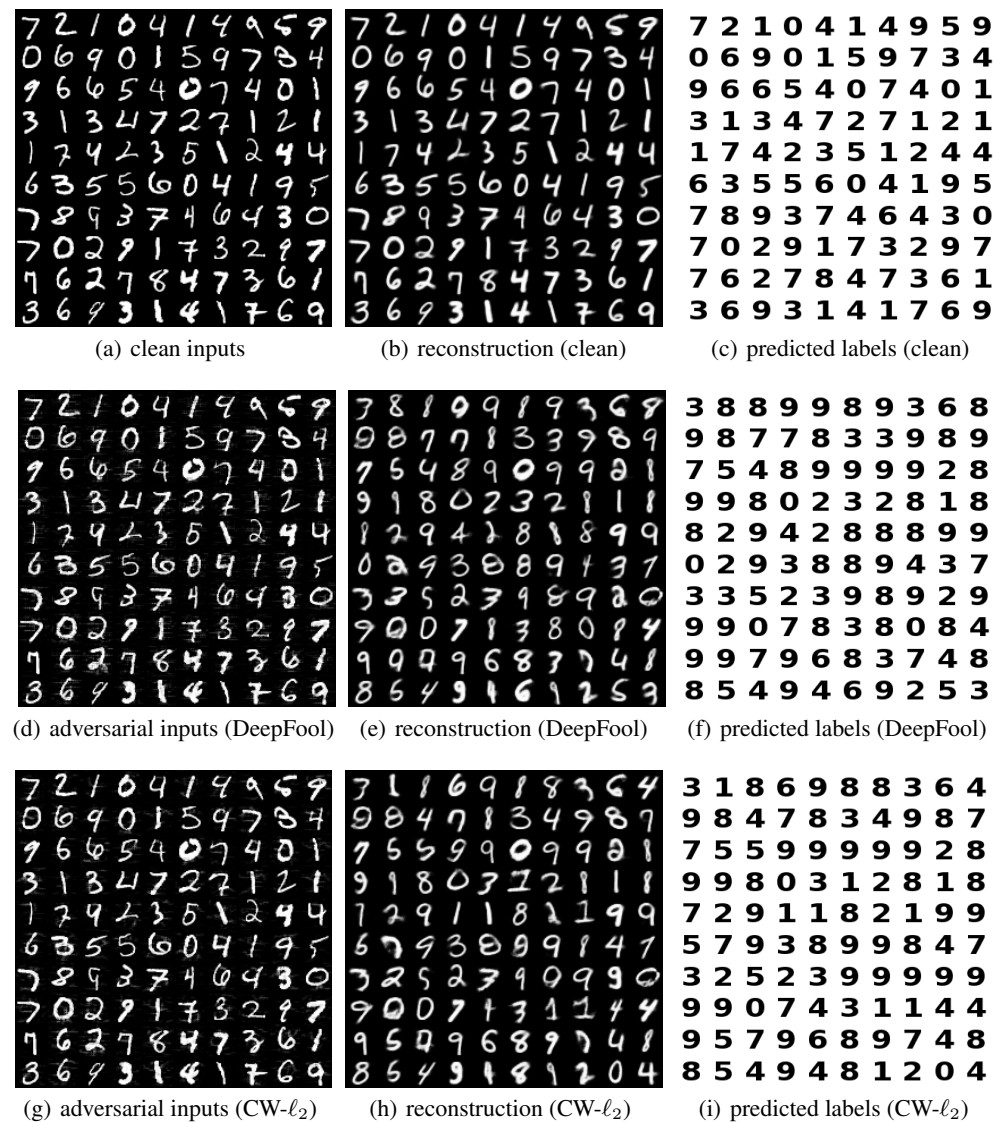

Figure 2: Visualising the clean, adversarial and reconstructed images, as well as the labels on the clean/adversarial inputs. Many of the reconstructed images from the adversarial inputs are visually more close to the predicted labels on the adversarial images.

## C  PARAMETERS OF THE ATTACKS

**CW-$\ell_2$:**  as recommended by `https://github.com/carlini/nn_robust_attacks/blob/master/l2_attack.py`

**DeepFool:**  as recommended by `https://github.com/tensorflow/cleverhans/blob/master/cleverhans/attacks.py#L1092`

