# OpenReview forum: "Are Generative Classifiers More Robust to Adversarial Attacks?"
_ICLR.cc/2018/Workshop — Reject_

### Official Review · AnonReviewer2 · 2018-03-09
**Anecdotal evidence on generative classifiers and adversarial examples**

**Rating:** 4
**Confidence:** 4

**Review:**

This paper finds that one class of deep generative neural networks is less vulnerable to the FGSM attack than a standard CNN on MNIST. Of course, this may be due to gradient masking (which the paper, to its credit, does acknowledge), and the advantage may or may not hold on other datasets or when the CNNs are trained robustly (e.g., with the methods of Madry et al. (2018)). There could be something interesting here -- it may indeed be the case that generative models tend to be more robust. However, the evidence is inconclusive. Before this paper is ready for publication (even as a workshop paper), there needs to be clearer evidence that the result is real. This is especially important for work on adversarial examples, since many papers have failed to live up to their claims when evaluated against stronger attacks.

---

### Official Review · AnonReviewer3 · 2018-03-10
**An empirical study of the effect of adversarial attacks on generative classifiers**

**Rating:** 6
**Confidence:** 2

**Review:**

The paper discusses some empirical results that seem to suggest that generative classifiers may be more robust to adversarial attacks than discriminative classifiers. I’m not an expert on the topic, but from my very limited knowledge the paper seems to be quite clear in the distinction with the related work.

The paper is well-written and the work seems novel enough, but not yet very well developed. If I’m not missing something, my impression is that it is relying mostly on limited empirical results on MNIST.

Pros:
- The topic is quite interesting and has significant applications.
- The paper is generally well-written.

Cons:
- This seems to be mostly a relatively limited empirical evaluation, so I'm not sure one could draw any major conclusions yet.

---

### Decision · Program_Chairs · 2018-03-20
**ICLR 2018 Workshop Acceptance Decision**

**Decision:**

Reject

**Comment:**

Based on the reviews, this paper has not been accepted for presentation at the ICLR workshop. However, the conversation and updates can continue to appear here on OpenReview.